# New nanofiber glaucoma drainage implant: Effectiveness, safety, first *in vivo* results, and optimization of surgical technique

Adela Klezlova[1], Petr Bulir[1], Alexandr Stepanov[1,2,3]*, Andrea Sidova[4],
Magdalena Netukova[1], Jana Vranova[1], Katarina Urbaniova[1], Martina Grajciarova[5,6],
Lenka Vankova[5], Zbynek Tonar[5,6], Pavel Studeny[1]

**1** Ophthalmology Department, Third Medical Faculty, Charles University and University Hospital Kralovske
Vinohrady, Prague, Czech Republic, **2** Ophthalmology Department, Klaudians Hospital, Mlada Boleslav,
Czech Republic, **3** Ophthalmology Department, Medical Faculty, Slovak Medical University in Bratislava,
Bratislava, Slovak Republic, **4** Department of Nonwovens and Nanofibrous Materials, Technical University
of Liberec, Faculty of Textile Engineering, Liberec, Czech Republic, **5** Department of Histology and
Embryology, Faculty of Medicine in Pilsen, Charles University, Prague, Czech Republic, **6** Biomedical
Center, Faculty of Medicine in Pilsen, Charles University, Prague, Czech Republic

* stepanov.doctor@gmail.com

## Abstract

### Purpose

The purpose of the study is to evaluate the effectiveness, surgical preoperative and
postoperative complications, histopathological findings, and optimize surgical tech-
nique after implantation of the new nanofiber glaucoma drainage implant (GDI).

### Method

Implantation of the GDI, a unique nanofiber drainage device fabricated from polyvi-
nylidene fluoride (PVDF) using the well-established electrospinning technology on the
Nanospider™ platform, was first optimized *in vitro* on cadaver porcine bulbs before
the initial *in vivo* implantations. PVDF was selected due to its favorable properties,
including biocompatibility, anti-adhesive behavior, and mechanical stability, which are
particularly advantageous in minimizing fibroblast colonization and fibrotic encapsu-
lation. The Nanospider™ technology allows for reproducible, large-scale fabrication
of nanofiber materials with controlled fiber morphology, which ensures uniformity and
precision of implant dimensions.
*An in vivo* study on 28 normotensive eyes from 14 laboratory New Zealand White
rabbits was conducted. There were two groups of animals: the study group (14 eyes)
and the control group (14 contralateral eyes). The study group underwent implan-
tation of the new nanofiber GDI; the control group did not undergo any surgical
procedure. Intraocular pressure (IOP) was measured preoperatively and at regular
times postoperatively (Tono-Pen AVIA®). Preoperative and immediate postoperative

journal.pone.0335858

Hospital, Henan University of Science and
Technology, CHINA

**Peer Review History:** PLOS recognizes the
benefits of transparency in the peer review
process; therefore, we enable the publication
of all of the content of peer review and
author responses alongside final, published
articles. The editorial history of this article is
available here: https://doi.org/10.1371/journal.
pone.0335858

**Data availability statement:** All data reported in the manuscript are publicly available: Open Science Framework, DOI 10.17605/OSF.IO/B9QM6.

**Funding:** The study received support from the Charles University Cooperatio Program, research area MED/DIAG, the grant SVV – 2025 No 260 773, and from the Ministry of Education, Youth and Sports under the project FIND No. CZ.02.1.01/0.0/0.0/16_019/0000787, UNCE/MED006 Center of Excellence (Charles University), and AZV, grant number: NU 23-08-00586. The funders had no role in study design, data collection and analysis, decision to publish, or preparation of the manuscript.

**Competing interests:** The authors have declared that no competing interests exist.

complications were monitored. Histological quantification was performed using unbiased sampling and stereological methods to assess leukocyte infiltration, type I and type III collagen fractions, and both absolute and relative levels of inflammation.

## Results

Based on the previous results and *in vitro* surgical experiences, the implant was narrowed to 2.0 mm, a thickness of 100 µm was chosen, and the implant was fixed with two scleral stitches to maintain its position. No serious preoperative complications occurred during *in vivo* experiments. There was one extrusion of the glaucoma implant noted after surgery, likely due to insufficient conjunctival fixation. This animal was excluded from both the study and the control groups. No serious instances of intraocular hypotension were observed after surgery. All animals tolerated the surgical procedure well, and the postoperative period was without any serious issues. In the study group, the average preoperative IOP was 13.6 mmHg (±4.1, n = 13). The average postoperative IOP on the first day, one, two, and three weeks, and one month after surgery decreased to 8.8 mmHg (±3.3, n = 13), 9.8 mmHg (±2.0, n = 13), 10.3 mmHg (±3.6, n = 13), 10.2 mmHg (±2.6, n = 13), and 9.7 mmHg (±2.0, n = 13), respectively. In the control group of contralateral eyes, the average preoperative IOP was 11.42 mmHg (±4.2, n = 13). The average postoperative IOP was 11.8 mmHg (±5.4, n = 13), 14.2 mmHg (±4.6, n = 13), 14.5 mmHg (±3.4, n = 13), 14.0 mmHg (±3.8, n = 13), and 14.2 mmHg (±2.4, n = 13), respectively, at the same follow-ups. In the study group, the IOP was statistically significantly lower by 29% at the end of the follow-up compared to the preoperative measurements ($p = 0.009$). Eyes with the implant showed greater leukocyte infiltration and less type I collagen compared to the group without implants. The ratio of type I to type III collagen was lower in the implant group, indicating delayed maturation and weaker connective tissue during early healing.

## Conclusion

For easier implantation, minor technical adjustments such as implant narrowing and scleral fixation of the GDI were developed and tested using *in vitro* experiments. *In vivo* implantation of unique nanofiber GDI appeared safe and technically well-suited for our study. No serious perioperative or postoperative complications were observed. There was one scleral extrusion of the device, which was, in our opinion, caused by insufficient conjunctival fixation. A statistically significant IOP reduction was achieved at the end of the follow-up in the study group with implanted GDIs. Further studies on the effectiveness of the implant with longer monitoring periods, together with other surgical options such as combined cataract surgery and nanofibers GDI, are needed.

## Introduction

Glaucoma is an irreversible, chronic neuropathy, causing thinning of the retinal nerve fiber layers, progressive optic nerve damage, and gradual loss of sight in both adults

and children [1]. Despite significant progress in diagnosis and treatment, this disease remains vision-threatening, and no long-term effective therapy is currently available. From the therapeutic and pathophysiological aspects, intraocular pressure (IOP) is a major modifiable risk factor in the development and progression of glaucoma disease, and reducing IOP is the primary goal of any treatment [2,3]. There are generally three options for achieving IOP reduction with conservative treatment: (i) local or oral medications, (ii) laser treatment, or (iii) incisional surgery. When medications or laser therapy are not sufficiently effective in reducing IOP, incisional surgery such as trabeculectomy (TE), glaucoma drainage implants (GDI), or nonpenetrating or microinvasive glaucoma surgery (MIGS) is necessary.

New biomaterials offer a broad range of applications in medicine and hold significant promise for glaucoma treatment [4]. Nanofibers provide several unique properties for the engineering of synthetic trabecular meshwork (TM) because their fabrication techniques can be modified to develop the architecture of the TM. The synthetic polymer polyvinylidene fluoride (PVDF) is particularly notable for its biocompatibility, non-degradability, non-toxicity, cell-growth resistance, and anti-adhesive properties, as consistently demonstrated in our long-term experimental work with PVDF-based nanofibrous materials. These findings further support its suitability for biomedical applications, including glaucoma surgery [5].

PVDF is widely used in medicine for hernia meshes, urogynecologic implants, non-resorbable surgical suture threads (also in eye surgery), etc. [6,7]. Our study is devoted to the verification and optimization of a unique PVDF-based nanofiber glaucoma implant fabricated using a well-established electrospinning technique. This technology (Nanospider™) is an established electrospinning method that enables large-scale production of nanofibrous materials with consistent quality and minimal variability. Unlike conventional needle-based electrospinning, the Nanospider™ device employs a free liquid surface as the polymer source, which allows the simultaneous formation of thousands of nanofibers under the influence of a strong electric field. This approach not only facilitates industrial-scale manufacturing but also ensures high uniformity and tunability of fiber morphology according to the requirements of specific applications, such as replicating the structural and functional characteristics of the TM. The required size of the implant was adjusted in the laboratory using an exact cutting method under microscopic guidance, ensuring maximal dimensional accuracy of the final product.

We evaluated the initial short-term outcomes, including preoperative and early postoperative complications, the tendency for aqueous humour drainage via subconjunctival filtration, and *in vivo* assessments of fibrosis and biocompatibility.

## Materials and methods

### Ethics statement

This study was carried out in strict accordance with law number 246/1992 Sb. and the Guide for the Care and Use of Laboratory Animals §15d CZ 04044 of the Czech Republic. The protocol was approved by the Ministry of Health of the Czech Republic (Protocol Number: 69–2022). All surgery was performed under sodium pentobarbital anesthesia, and all efforts were made to minimize suffering.

### Material development, *in vitro* qualities

Nanofiber materials were fabricated using PVDF (Mw ≈ 180.000 g/mol; Sigma-Aldrich, Germany) as the base polymer. The electrospinning solution was prepared by dissolving PVDF in a binary solvent system consisting of dimethylacetamide (DMAC; Penta Chemicals) and acetone (AC; Penta Chemicals) in a volume ratio of 4:1, yielding a final polymer concentration of 26 wt.%. The dissolution process was carried out at 47 °C for 24 hours under continuous stirring. A detailed protocol for the preparation of the polymer solution is described by Klapstova et al. [5]. Electrospinning of the prepared solution was performed using a needleless electrospinning device (Nanospider 1WS500U; Elmarco, Czech Republic), schematically illustrated in Fig 1. The processing parameters were systematically optimized based on preliminary studies with PVDF-based systems and are summarized in Table 1.

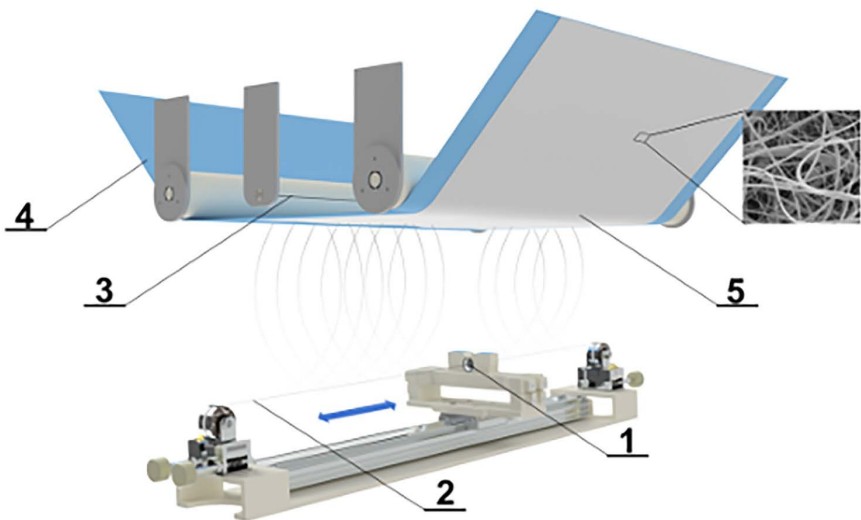

**Fig 1. Scheme of needleless electrospinning process.** 1) The polymer solution applied through a steel die, 2) wire electrode with a positive electrical voltage, 3) the counter electrode, 4) polypropylene substrate textile, type spunbond, 5) the nanofibers.

**Table 1. Electrospinning conditions applied for the fabrication of nanofibrous PVDF layers.**

| Distance between electrodes [mm] | 180 |
|---|---|
| High voltage on electrodes (counter/wire) [kV] | −10/+40 |
| Rewinding speed [mm/min] | 20 |
| Steel orifice diameter [mm] | 0.7 |
| Temperature [°C] | 22 |
| Relative air humidity [%] | 60 |

## Material characterization

The fabricated nanofiber materials were systematically characterized with respect to thickness, area weight, and surface morphology. Thickness was determined using a digital thickness gauge (Elcometer 456, Gamin s.r.o., Czech Republic) at ten randomly selected locations. The area weight was assessed by excising circular specimens with a surface area of 100 cm² from the material, followed by precise gravimetric analysis, also in ten replicates. Morphological evaluation was performed using scanning electron microscopy (SEM; Tescan Vega 3SB Easy Probe, Czech Republic). Quantitative analysis of fiber diameter was conducted using NIS Elements software (LIM s.r.o., Czech Republic), based on a dataset comprising 500 individual measurements obtained from representative SEM micrographs.

## *In vitro* characterization

The biological performance of the fabricated scaffolds was evaluated using 3T3 (SA) mouse fibroblast cell lines (ATCC, USA). Before cell seeding, the scaffolds were cut into circular specimens with a diameter of 15.5 mm and subsequently sterilized with ethylene oxide at 21 °C for 12 hours. Following sterilization, the samples were aerated for a minimum of 7 days to ensure the removal of residual sterilization agents. The 3T3 fibroblasts were maintained in Dulbecco's Modified Eagle Medium (DMEM; Lonza Biotec s.r.o., Czech Republic) supplemented with 10% fetal bovine serum (FBS; Lonza Biotec s.r.o., Czech Republic), 1% glutamine (Biosera, Czech Republic), and 1% antibiotic-antimycotic solution (penicillin/streptomycin/amphotericin B; Lonza Biotec s.r.o., Czech Republic).

Cells were cultured in a humidified atmosphere containing 5% $CO_2$ at 37 °C. For seeding, fibroblasts from passage nine were applied onto the scaffolds at a density of $1 \times 10^4$ cells per well. Cell-material interactions were assessed at 1- and 8-days post-seeding using an MTT (3-[4,5-dimethylthiazol-2-yl]-2,5-difenyltetrazolium bromide) assay, following established protocols. After incubation, the optical density of the resulting suspension was measured ($\lambda$sample = 570 nm, $\lambda$reference = 650 nm). The resulting absorbance values were determined spectrophotometrically (Spark, Tecan, Switzerland).

Additionally, automated quantification of stained cell nuclei was performed on ten randomly selected fields of view using an inverted fluorescence microscope (Nikon Eclipse Ti-E; Nikon Imaging, Czech Republic) and MATLAB software. Results expressed per unit area (1 mm²). The samples were washed twice with PBS and subsequently fixed in 2.5% glutaraldehyde for 30 minutes. After additional double washes with PBS, cell nuclei were stained with 4′,6-diamidino-2-phenylindole (DAPI; 1:1000 dilution; Merck KGaA, Germany) following standard procedures. Cell nuclei were automatically counted in MATLAB by first isolating the blue channel from the DAPI-stained images. The background was removed using a top-hat transform, followed by global thresholding (T = 0.2) to generate a binary image. Small artifacts were subsequently eliminated through morphological opening. Connected nuclei were separated using watershed segmentation, and the final counts per unit area were calculated from the original image size and plotted with 95% confidence intervals. The methodology for cell quantification follows the approach described by Horakova et al. [8].

### *In vitro* optimization of surgical technique

Before the *in vivo* study, a method of implantation of the GDI was tested on porcine cadaver eyeballs. Due to its comparable size and anatomical structure, the porcine eye serves as an optimal model for refining implantation techniques and enhancing surgical training for subconjunctival glaucoma drainage implant (GDI) procedures [9]. This step helped to estimate the behavior of the material *in vivo*, optimize the surgical procedure, and reduce the number of animals needed for the *in vivo* study.

The procedure began with the dissection and mobilization of the conjunctiva from the underlying sclera, extending approximately 10 mm along the limbus of the cornea to expose the surgical site. A calibrated microknife was used to create a scleral incision 3 mm posterior to the corneal limbus, measuring 2.2 mm in width and 0.3 mm in depth. A scleral tunnel was then fashioned using a bevel-up knife, extending 0.5 mm into the clear cornea. Entry into the anterior chamber was achieved with a trapezoidal 2.2 mm knife. A planar nanofiber GDI with a size of 2.2 x 6 mm was inserted into the prepared scleral tunnel using a spatula. The implant was placed very close to the edge of the anterior chamber entrance (Fig 2). The distal part (the part that extends under the conjunctival flap) was bent back to the beginning of the scleral incision and fixed by two individual 11/0 polypropylene sutures. The sutures were used to prevent implant dislocation into the anterior chamber or vice versa, i.e., under the conjunctiva. Finally, the conjunctiva was fixed using a continuous resorbable suture.

During the *in vitro* phase, we made several modifications. The originally planned width of the GDI was reduced from 2.2 to 2 mm, which proved to be much easier for insertion into a 2.2 mm wide scleral tunnel. After several surgical tries using implants of three different thicknesses (50, 100, 150 μm), we decided on the 100 μm, which was still easily surgically implantable and proved to have good filtration properties in *in vitro* analysis. Implants thinner than 100 μm were prone to tearing, while thicker variants proved difficult to insert into the sclera. The 100 μm implant was later verified using *in vivo* rabbit eyes. A schematic representation of the surgical procedure and GDI dimensions are shown in Fig 3.

### Methods of the in vivo study

Fourteen laboratory-bred New Zealand White rabbits were used as an animal model for the *in vivo* study. Surgical procedures were performed on one normotensive eye per animal, without induction of glaucoma. Eyes were divided into two groups: the Study group (14 eyes) and the Control group (14 contralateral eyes). The study group underwent implantation

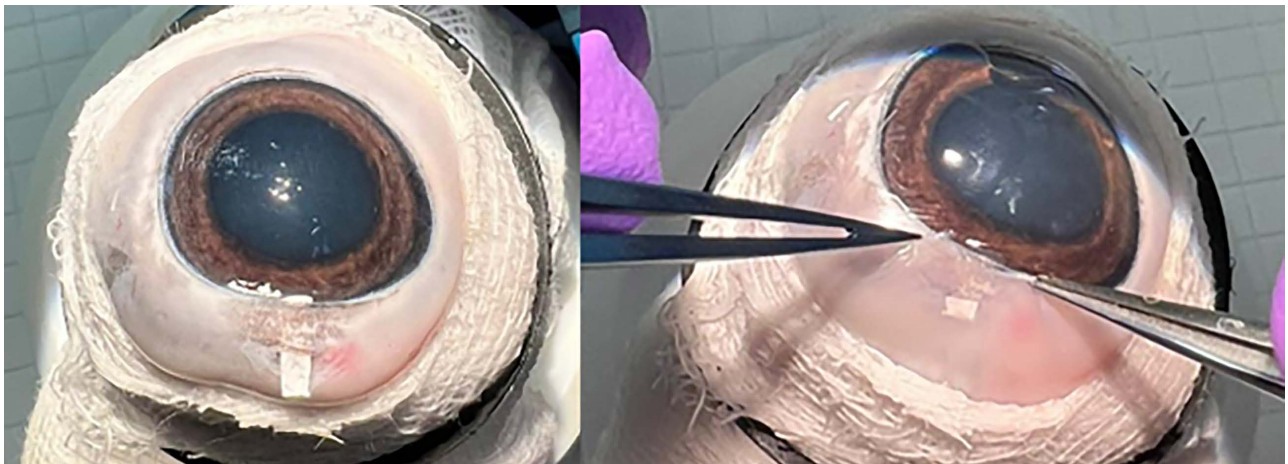

**Fig 2. A** In vitro GDI inserted via scleral tunnel **B** GDI bent and covered by the conjunctiva.

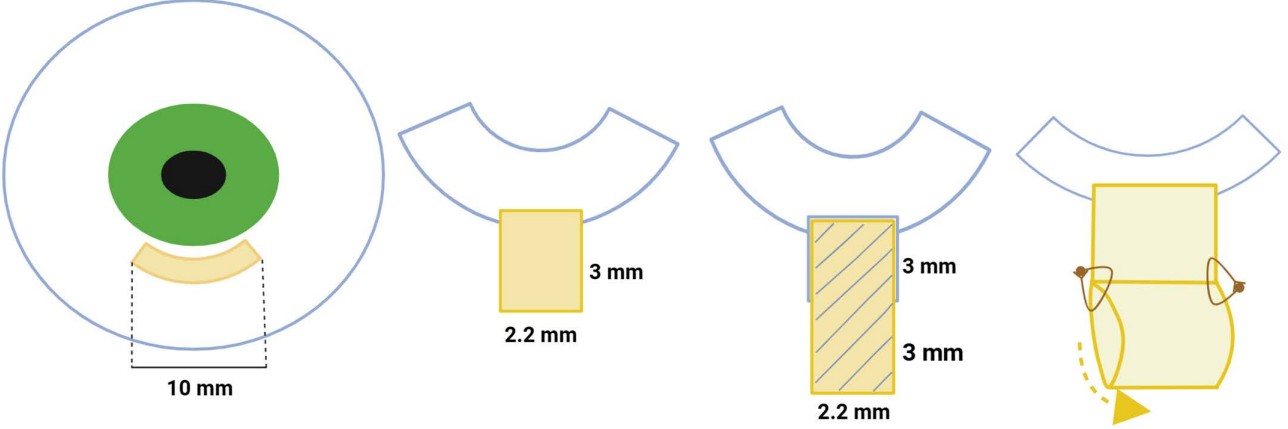

**Fig 3. Scheme of the surgical procedure.** Preparation of conjunctiva, scleral incision, insertion of the implant, bending and fixation of GDI.

of the new nanofiber GDIs, while the control group did not undergo any surgical procedure. All surgeries were carried out under general anesthesia. The surgery was initiated after disinfection of the operating field (povidone iodine solution) and preparation of the area from the conjunctiva to the sclera, along the limbus of the cornea to a distance of about 10 mm. Any minor bleeding was stopped by compression. The next steps of the surgery continued according to the *in vitro* procedure (part number 3) including insertion of the GDI. Postoperatively, a combined antibiotic and corticosteroid ointment containing tobramycin (3 mg/g) and dexamethasone (1 mg/g) was applied once daily for a minimum of four days following surgery.

Intraocular pressure (IOP) was measured preoperatively (prior to general anesthesia) and at regular intervals postoperatively, on the first day, and at one, two, and three weeks, as well as one month, using the Tono-Pen AVIA®. Measurements were taken under topical anesthesia with 0.4% oxybuprocaine hydrochloride in both eyes.

The postoperative finding of the operated eyes was photo-documented using a microscope at each IOP measurement.

All animals were monitored for a period of one month. Animals were assessed daily, including their general health condition, behavior, and food and water intake. A veterinary specialist provided general postoperative care. At the conclusion

of the study, euthanasia was performed in accordance with all ethical guidelines. Prior to sample collection, the animals were subjected to general anesthesia via intramuscular administration of ketamine and medetomidine, and subsequently euthanized by intravenous administration of pentobarbital. The eyes were enucleated and sent for histopathological analysis.

## Histological processing and quantification

In the histological phase, two groups of rabbit eyes were evaluated: the study histological group (7 eyes) and the control histological group (7 eyes). The study histological group consisted of rabbit eyes with implants extending into the area of the conjunctiva, sclera, iridocorneal angle, and ciliary body; the control histological group was created from the intact parts of the same eyes (serving as internal controls) (Fig 4).

Tissue samples were fixed in a 10% neutral buffered formalin solution for two weeks, dehydrated in ethanol, embedded in paraffin blocks, and cut into 5 µm-thick sections (Leica RM2255 microtome). The samples were cut into 100 µm-long segments in the ventrodorsal direction. Histological sections were then deparaffinized, rehydrated, and stained using three histological staining methods: hematoxylin-eosin [10], Verhoeff´s hematoxylin and green trichrome [11], and picrosirius red [12].

The histological quantification involved assessing area fractions of leukocytes, total collagen, and collagen types I and III. Additionally, both the absolute and relative thickness of inflammatory tissue were measured.

Stereological measurements of area fractions were performed using Stereologer 11 software (SRC Biosciences, Tampa, FL, USA), in combination with an Olympus BX41 microscope (Japan) with a standard set of Plan Fluor objectives, a Promicra 3−3CC camera (Olympus, Japan), and a ProScan III motorized 3-axis step motor (Prior Electronics, UK). Histological measurements were conducted in real-time using live imaging. The sampling strategy was based on systematic and uniform sampling of rabbit eyes [13]. Sampling was used for each of the segments. Area fractions were evaluated using a stereological point grid and the Cavalieri principle using a 40x objective (Olympus Optical Co., Ltd., Tokyo, Japan) (Fig 5A-5D) [14], as described in previously published studies [15,16]. All fractions were calculated as the ratio of the structure of interest to the reference area.

Stereological evaluations of the absolute and relative thickness of inflammation were performed using Ellipse software (ViDito, Košice, Slovak Republic). One photomicrograph per sample was taken for quantification of inflammatory thickness using a 4x objective (Olympus Optical Co., Ltd., Tokyo, Japan), resulting in a total of 103 images. Absolute thickness of

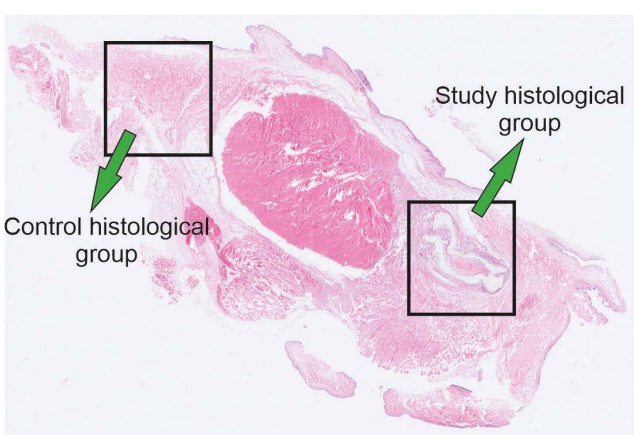

**Fig 4. Groups of the histological analysis.** The implant group consisted of rabbit eyes with implants. The without implant group was obtained from the intact part of the same eye.

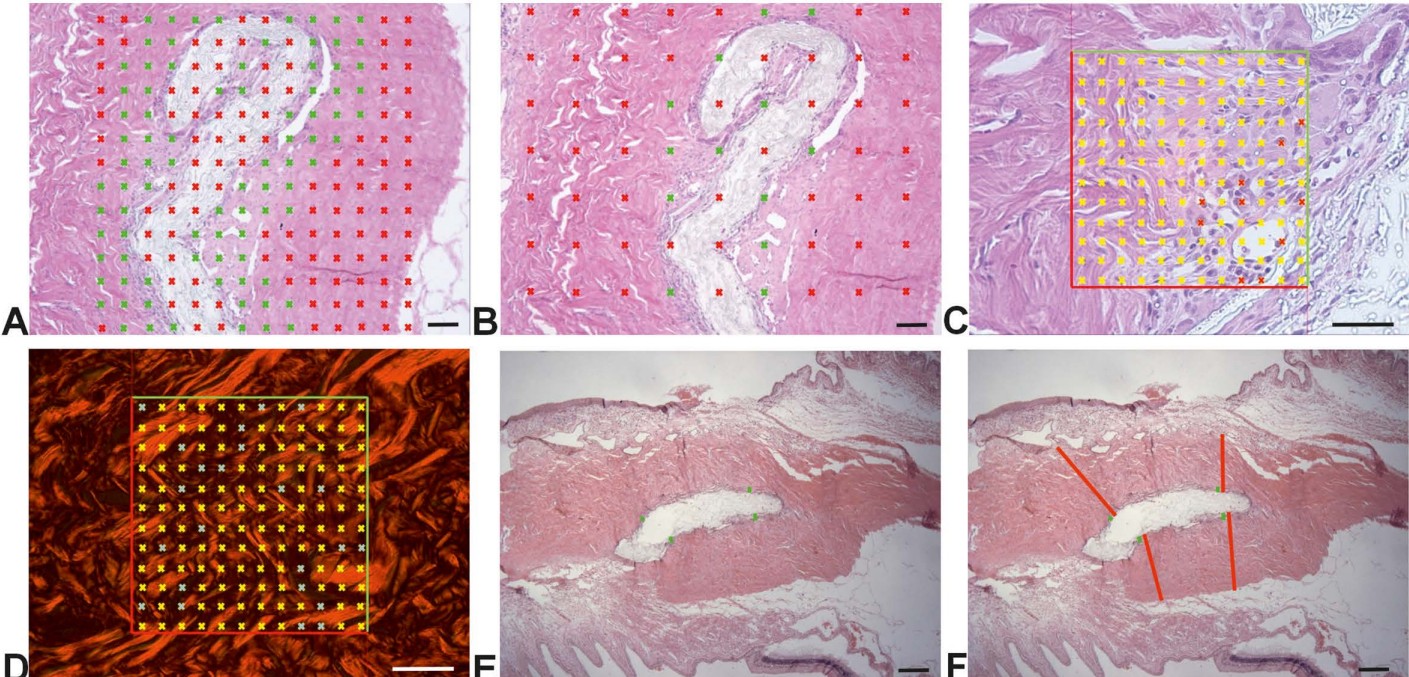

**Fig 5. Systematic uniform sampling of the rabbit eye and quantification of a fraction of microscopic structures using Stereologer 11 software (A-D) and inflammatory thickness using Ellipse software (E-F).** A – The region volume for evaluating of the fraction of leukocytes and collagen. B – The selection of the series of fields of view using the Cavalieri principle. C – The fraction of leukocytes was evaluated using a stereological point grid and the Cavalieri principle. D – The fraction of type I and III collagen was evaluated using a stereological point grid and the Cavalieri principle. E – The absolute inflammatory thickness was quantified as a value of measurements by four linear probes (green color). F – The relatively inflammatory thickness was evaluated as the ratio of the average distance of leukocytes (green color of linear probes) from the implant to the distance from the edge of the part of the eye in which the implant is located (red color of linear probe). Scale bars 100 µm **(A, B)**, 50 µm **(C, D)**, 500 µm **(E, F)**.

inflammation was quantified by averaging four linear measurements taken from each photomicrograph using linear probes (Fig 5E). Relative thickness of inflammation was calculated as the ratio between the average radial distance of leukocytes around the implant and the total radial distance from the implant to the outer boundary of the surrounding anatomical region (Fig 5F).

## Statistical analysis

Intraocular pressure (IOP) values for eyes in the study and control groups were recorded at six time points: preoperatively, and on the first day, and at one, two, and three weeks, as well as one month postoperatively. Data are presented as means±standard deviations. Since IOP values at all time points met the assumption of normal distribution, parametric statistical tests were applied.

The independent Student's *t*-test was also used to identify statistically significant differences between the study and control groups in terms of IOP change from the preoperative baseline to each postoperative time point (i.e., first day, and one, two, and three weeks, and one month after surgery). Additionally, a repeated-measures analysis of variance (ANOVA) was performed separately for each group to evaluate statistically significant differences in IOP across the six observed time intervals. Statistica version 14.0.0.15 (TIBCO Software Inc., CA, USA) was used for statistical analysis. A *p*-value of less than 0.05 was considered to be statistically significant.

Histological data were analyzed separately. Since some data did not pass the Shapiro–Wilk test for normality, nonparametric methods were used. Both groups were compared using the Wilcoxon matched-pairs test. These statistical analyses were performed using the Statistica Base 11 package (StatSoft, Inc., Tulsa, OK, USA).

## Results

### Material characterization

The surface morphology of the electrospun PVDF scaffolds was characterized using SEM. As illustrated in Fig 6, the nanofiber structure exhibited a uniform fiber distribution without the presence of significant bead defects, indicating stable electrospinning conditions. The fibers displayed a smooth morphology with an average diameter of 365±38 µm, as determined from a representative set of measurements. The fabricated nanofiber layer exhibited an area weight of approximately 20 g/m², corresponding to a thickness of around 100 µm. The measured thickness was identified as optimal based on previous experimental findings, providing a suitable balance between mechanical robustness and handling characteristics required for surgical manipulation. The porosity of the GDI was determined in accordance with ASTM F316 using a capillary flow porometer (Porometer 3G, Quantachrome, USA). The measured porosity was 85.62±2.85%.

### *In vitro* characterization

The aim of the *in vitro* study with 3T3 mouse fibroblasts was to verify whether the material had the required anti-fibrotic properties without cytotoxic effects. If cells did not grow on the material, the material was considered suitable as a GDI. The colorimetric MTT test was applied after 1 and 8 days of cultivation, and measured cell viability as well as cell quantification. Results are shown in Fig 7.

As can be seen in Fig 7, after 8 days of cultivation, there was a slight increase in cell population; however, the absorbance value was still very low. The same effect was observed in cell quantification, where the number of measured cells increased from 88 after the first day to 175 after 8 days of cultivation.

Nevertheless, the overall absorbance values remained markedly low, indicating limited cell viability and metabolic activity on the material surface. These findings were supported by the cell quantification analysis, where the number of detected cells increased from 88±42 cells on day 1–175±61 cells after 8 days. The minimal cellular attachment and proliferation suggest that the material effectively inhibits fibroblast colonization, a critical requirement for minimizing fibrotic

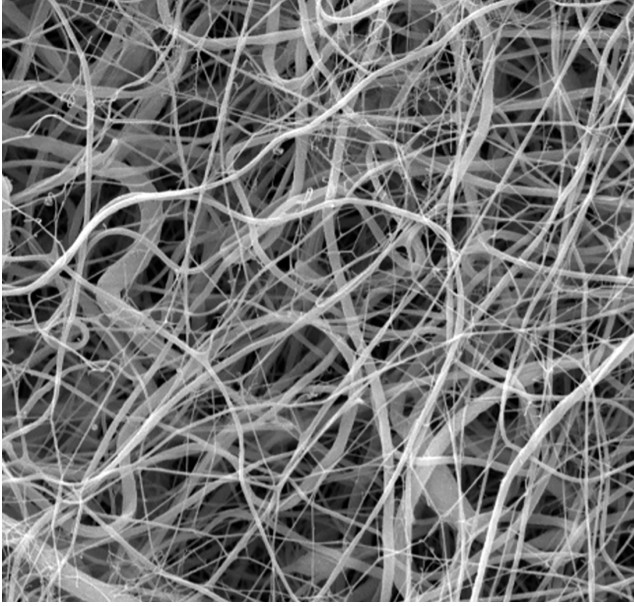

**Fig 6. SEM image of the electrospun PVDF material, scale bar 10 µm, magnification 5.000x.**

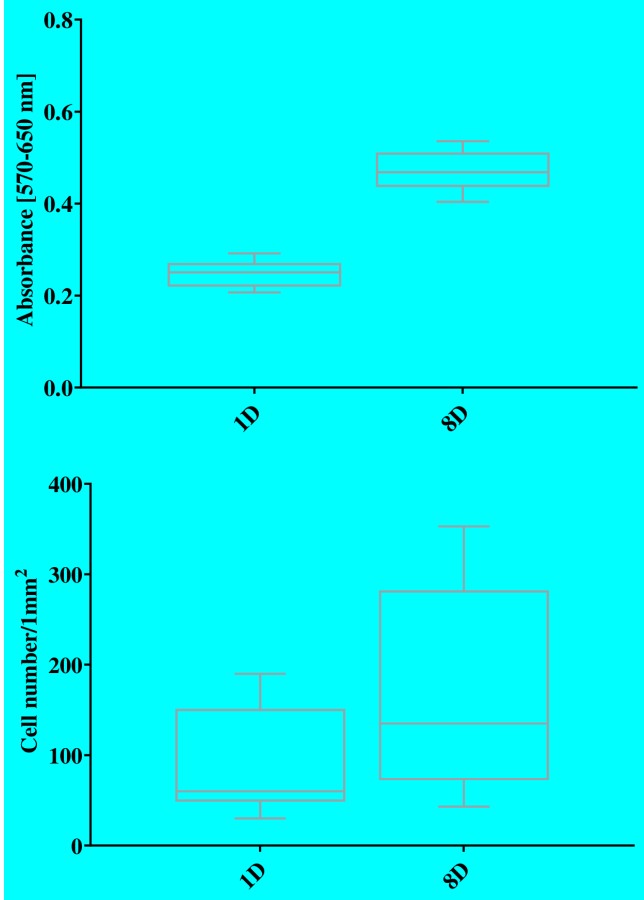

**Fig 7. Results of the colorimetric MTT test.**

encapsulation in glaucoma drainage systems. Furthermore, the absence of rapid cell death supports the conclusion that the material is non-cytotoxic and biocompatible under the tested conditions.

### Preoperative and postoperative complications

We did not observe any serious preoperative or postoperative complications. There was one extrusion of the glaucoma implant noted eleven days after the surgery, which was probably caused by insufficient conjunctival fixation. This animal was excluded from statistical analyses. Furthermore, no instances of intraocular hypotony were observed throughout the entire follow-up period. Migration of the glaucoma drainage implant (GDI) into the anterior chamber was not detected at any point during the study. Postoperative examinations revealed no cases of intraocular infection or hyphema. All animals tolerated both the surgical procedure and the postoperative period well, without any serious complications.

### IOP evaluation

In our study group, the average preoperative IOP was 13.6 mmHg (±4.1, n = 13). Postoperatively, IOP decreased to 8.8 mmHg (±3.3, n = 13), 9.8 mmHg (±2.0, n = 13), 10.3 mmHg (±3.6, n = 13), 10.2 mmHg (±2.6, n = 13), and 9.7 mmHg (±2.0, n = 13) on day 1, week 1, week 2, week 3, and month 1, respectively. In the control group the average preoperative IOP

was 11.42 mmHg (±4.2, n = 13) with postoperative values of 11.8 mmHg (±5.4, n = 13), 14.2 mmHg (±4.6, n = 13), 14.5 mmHg (±3.4, n = 13), 14.0 mmHg (±3.8, n = 13), and 14.2 mmHg (±2.4, n = 13) at the same time points.

The lowest IOP was observed on the first day after surgery, although no cases of postoperative hypotony were observed. In the study group, the IOP was statistically significantly lower by 28.9% at the end of the follow-up in comparison to the preoperative measurements ($p = 0.009$). Mean baseline and post-operative IOP measurements are listed and shown in Table 2 and Fig 8.

### Quantitative histological analysis

Complete data of the comparison of the study histological and control histological groups are summarized in Table 3, and in Figs 9 and 10.

The study histological group had a greater fraction of leukocytes in all eye layers than the control histological group (Wilcoxon matched pairs test, *p = 0.007686* for conjunctiva, *p = 0.000655* for sclera, *p = 0.007686* for iridocorneal angle, *p = 0.043115* for ciliary body). The fractions of total collagen (*p = 0.027709* for conjunctiva, *p = 0.002162* for sclera, *p = 0.038153* for iridocorneal angle, *p = 0.043115* for ciliary body) and type I collagen (*p = 0.027709* for conjunctiva, *p = 0.008374* for sclera, *p = 0.038153* for iridocorneal angle) were smaller in the study histological group compared to the control histological group. However, the conjunctiva of the study histological group had a greater fraction of type III collagen (*p = 0.046400*) compared to the control histological group; in the other layers of the eye, this fraction was the same in both groups. The ratio of collagen I to III in the conjunctiva was greater in the control group (*p = 0.027709*).

### Qualitative histological analysis of the study histological group

A greater number of newly formed blood vessels were observed in the loose connective tissue of the conjunctiva (Fig 11A) and ciliary body in the implant (study) group. In the dense bundles of collagen fibers of the sclera, the number of newly formed blood vessels was smaller. A chronic inflammatory reaction was present in all layers of the eye around the implant. The greatest inflammatory reaction occurred in the conjunctiva (Figs 11A and 11B). Lymphocytic infiltration was predominant across all layers of the eye (Fig 11A). Numerous activated macrophages were observed near blood vessels, actively phagocytosing the implant material (Fig 11A). Some cells appeared to be multinucleated giant cells and contained parts of the implant (Figs 11B, 11C). Newly formed collagen fibers, especially type III, were seen around the implant (Figs 11D and 11E). Changes in collagen orientation and fibrotization were monitored.

## Discussion

There remains a clinical need for GDIs that minimize post-operative complications, reduce dependence on secondary surgical interventions, and enhance long-term IOP control. In this study, we present a novel and adaptable nano-structured GDI. Several device configurations, which were designed based on hydrodynamic principles, were evaluated to ensure long-term IOP reduction while mitigating the risk of post-operative hypotony. The nano-structured shunts exhibited

**Table 2. IOP fluctuation.**

| Time of measurement | Study group IOP (mmHg) | Control group IOP (mmHg) |
|---|---|---|
| Preoperative | 13,6 | 11.4 |
| 1 day after the surgery | 8,8 | 11.8 |
| 1 week after the surgery | 9,8 | 14,2 |
| 2 weeks after the surgery | 10,3 | 14.5 |
| 3 weeks after the surgery | 10,2 | 14.0 |
| 1 month after the surgery | 9,7 | 14.2 |

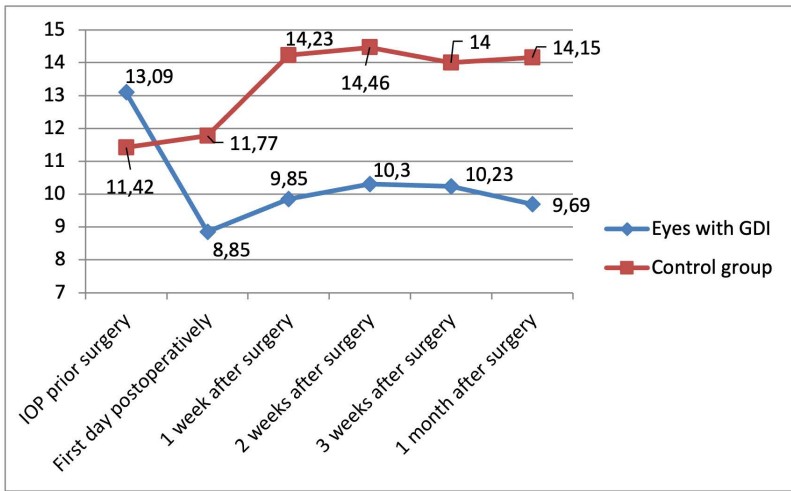

**Fig 8. IOP fluctuation.**

*in vitro* fluid dynamics, demonstrated leak-proof performance, and possessed sufficient mechanical integrity, i.e., strength, flexibility, and durability, needed for successful implantation. Our nano-structured GDI showed itself to be biocompatible, maintained patency, and significantly lowered IOP during the entire 4-week follow-up in a normotensive rabbit model without causing hypotony.

Although the long-term efficacy of HPE-designed MIGS devices in humans is still under investigation, our findings using nano-structured glaucoma drainage implants (GDIs) in healthy, normotensive rabbits demonstrate promising outcomes. Our GDI device, implanted without the use of mitomycin C (MMC), significantly reduced IOP during the 4-week follow-up compared to non-operated contralateral eyes. This contrasts with previous studies of TE and drainage implants in healthy, normotensive New Zealand White rabbits, where trabeculectomy (TE) procedures failed within seven days when performed without the use of antifibrotic or antimetabolic agents such as MMC [17].

There is a direct relationship between corneal endothelial cells (CEC) loss and the placement of the GDI. It was found that positioning the implant in the anterior chamber, very close or in direct contact with the corneal endothelium, caused progressive endothelial damage, which is significant since CEC has a very limited regenerative capacity. These issues are attributed to the stiffness of the implant, which prevents it from conforming to the curvature of the sclera [18]. For this reason, it is critical that implants be made from softer and more flexible materials. MIGS devices are evolving rapidly, with continuous improvements in design and performance. However, they lack the decades of development, clinical validation, and long-term data, i.e., over 40 years of evidence supporting visual field preservation and sustained intraocular pressure (IOP) reduction, that traditional glaucoma surgeries and implants can offer. Despite the progress in glaucoma surgical treatment, TE is the most frequently performed surgery in patients with advanced stages of glaucoma [19,20]. However, there are numerous potential complications, including scar formation (that can cause fibrosis), poor wound healing, hypotony, conjunctival erosions, bleb-associated infection, choroidal effusions, etc. [21].

Parikh et al. evaluated new versions of the nano-structured GDIs using closed-lumen and open-lumen systems [22].

Parikh et al. evaluated nano-structured glaucoma drainage implants (GDIs) featuring small-lumen designs, with and without a degradable inner core, to modulate aqueous humor outflow and reduce intraocular pressure (IOP) in normotensive rabbits [22].

The lowest average IOP was recorded 24 hours post-surgery in both open-lumen (8.0 ± 1 mmHg) and closed-lumen (13 ± 1 mmHg) implanted rabbit eyes, suggesting that the surgical procedure itself may transiently reduce IOP.

**Table 3. Differences between the implant and without implant groups. Data are displayed as the means and standard deviations (SD).**

| Quantitative parameter (unit) | Implant (mean ± SD) | Without implant group (mean ± SD) |
|---|---|---|
| $A_A$ (leukocytes, conjunctiva) (-) | greater (**) 0.12 ± 0.06 | smaller (**) 0.001 ± 0.001 |
| $A_A$ (leukocytes, sclera) (-) | greater (***) 0.04 ± 0.02 | smaller (***) 0.000 ± 0.000 |
| $A_A$ (leukocytes, iridocorneal angle) (-) | greater (**) 0.02 ± 0.01 | smaller (**) 0.000 ± 0.001 |
| $A_A$ (leukocytes, ciliary body) (-) | greater (*) 0.02 ± 0.01 | smaller (*) 0.000 ± 0.001 |
| $A_A$ (total collagen, conjunctiva) (-) | smaller (*) 0.55 ± 0.08 | greater (*) 0.76 ± 0.08 |
| $A_A$ (total collagen, sclera) (-) | smaller (**) 0.78 ± 0.07 | greater (**) 0.85 ± 0.04 |
| $A_A$ (total collagen, iridocorneal angle) (-) | smaller (*) 0.60 ± 0.09 | greater (*) 0.67 ± 0.05 |
| $A_A$ (total collagen, ciliary body) (-) | greater (*) 0.59 ± 0.02 | smaller (*) 0.48 ± 0.04 |
| $A_A$ (type I collagen, conjunctiva) (-) | smaller (*) 0.44 ± 0.10 | greater (*) 0.72 ± 0.09 |
| $A_A$ (type I collagen, sclera) (-) | smaller (**) 0.74 ± 0.07 | greater (**) 0.79 ± 0.04 |
| $A_A$ (type I collagen, iridocorneal angle) (-) | smaller (*) 0.53 ± 0.09 | greater (*) 0.60 ± 0.05 |
| $A_A$ (type I collagen, ciliary body) (-) | n.s. 0.47 ± 0.04 | n.s. 0.38 ± 0.05 |
| $A_A$ (type III collagen, conjunctiva) (-) | greater (*) 0.11 ± 0.03 | smaller (*) 0.05 ± 0.02 |
| $A_A$ (type III collagen, sclera) (-) | n.s. 0.04 ± 0.02 | n.s. 0.05 ± 0.02 |
| $A_A$ (type III collagen, iridocorneal angle) (-) | n.s. 0.08 ± 0.02 | n.s. 0.07 ± 0.02 |
| $A_A$ (type III collagen, ciliary body) (-) | n.s. 0.11 ± 0.02 | n.s. 0.10 ± 0.02 |
| Ratio of collagen I to III (conjunctiva) (%) | smaller (*) 5 ± 3 | greater (*) 17 ± 6 |
| Ratio of collagen I to III (sclera) (%) | n.s. 20 ± 9 | n.s. 20 ± 12 |
| Ratio of collagen I to III (iridocorneal angle) (%) | n.s. 7 ± 2 | n.s. 9 ± 2 |
| Ratio of collagen I to III (ciliary body) (%) | n.s. 4 ± 1 | n.s. 4 ± 1 |

The p values of the Wilcoxon matched pairs tests are shown * for $p < 0.05$, ** for $p < 0.01$, *** for $p < 0.001$, not statistically significant (n.s.) for $p > 0.05$.

Despite the initial drop in IOP, eyes implanted with the non-degradable core design showed no statistically significant difference compared to contralateral control eyes (16 ± 1 vs. 17 ± 1 mmHg, p = 0.06), suggesting that the surgical procedure alone does not result in sustained IOP reduction. No signs of hypotony or anterior chamber flattening were observed at any postoperative stage. IOP reduction in eyes with the non-degradable implant ranged from 4% to 25%, with the greatest decrease occurring immediately after surgery [22].

The eyes with open-lumen implant showed a statistically significant and sustained reduction in IOP compared to both control eyes (11 ± 1 vs. 16 ± 1 mmHg, $p < 0.0001$) and closed-lumen implant eyes (11 ± 1 vs. 16 ± 1 mmHg, $p < 0.0001$), with daily reductions ranging from 24% to 57% [22].

In our study, we recorded the lowest average IOP 24 hours post-surgery in the study group after nanofiber GDI implantation (8.8 ± 3.3 mmHg; $p < 0.0001$) with a daily reduction of 35.3%. No signs of hypotony or anterior chamber flattening were observed at any postoperative stage. At the end of the one-month follow-up, we noted a statistically significant decrease in IOP of 29% in comparison to the preoperative measurements (p = 0.009). Our results are thus comparable to those of the authors Parikh et al.

Eyes implanted with the degradable-core design demonstrated a statistically significant and sustained reduction in intraocular pressure (IOP) compared to both contralateral control eyes (11 ± 1 vs. 16 ± 1 mmHg, p < 0.0001) and non-degradable-core implants (11 ± 1 vs. 16 ± 1 mmHg, p < 0.0001), with daily IOP reductions ranging from 24% to 57% [22]. In our study, the lowest average IOP was recorded 24 hours post-surgery in the study group following nanofiber GDI implantation (8.8 ± 3.3 mmHg, p < 0.0001), corresponding to a 35.3% reduction. No signs of hypotony or anterior chamber flattening were observed at any postoperative stage. At the end of the one-month follow-up, we observed a statistically significant IOP decrease of 29% compared to preoperative values (p = 0.009). These results are comparable to those reported by Parikh et al.

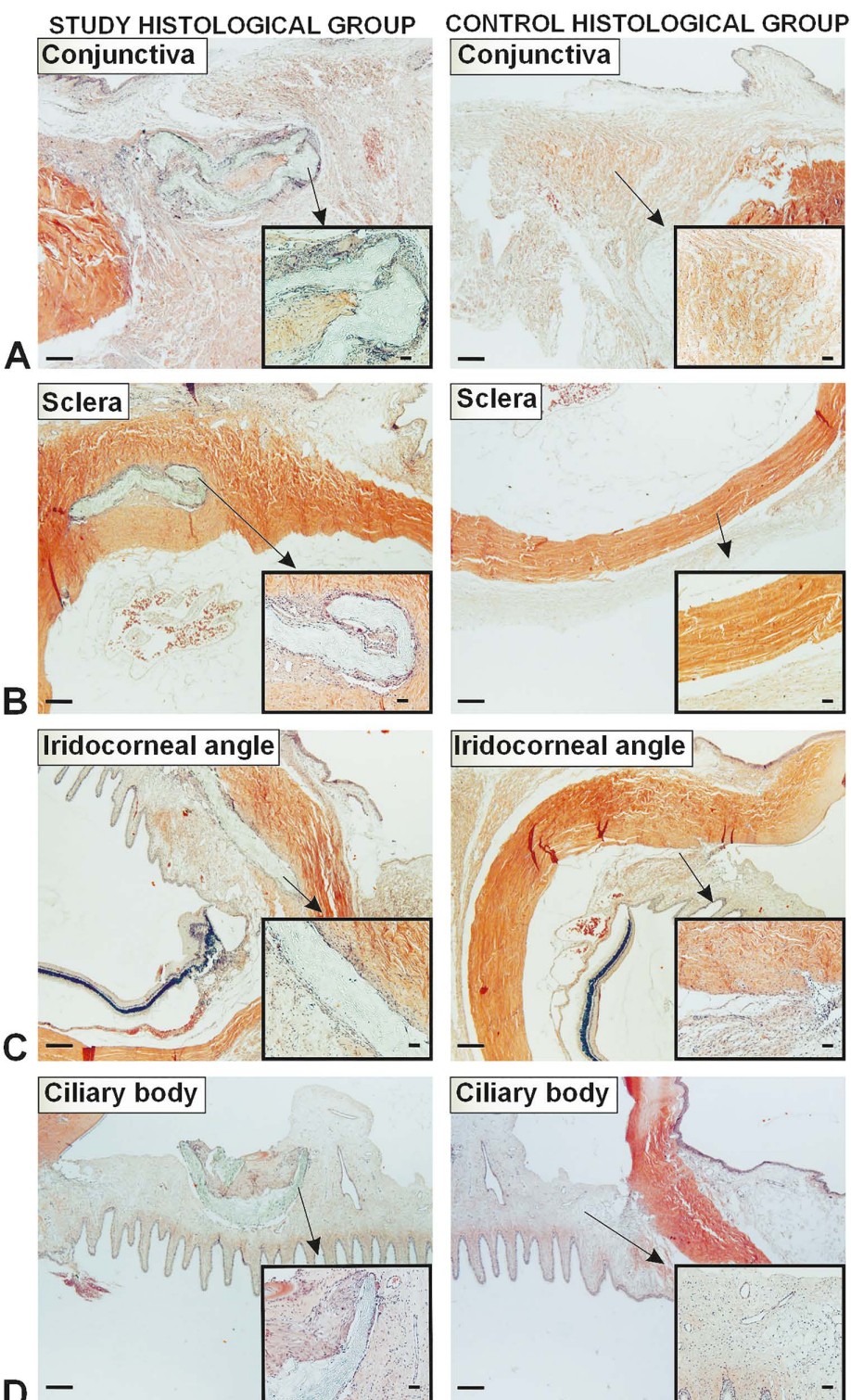

**Fig 9. Comparison of leukocyte infiltration in the implant and without implant groups.** Scale bars 500 μm (A-D, left side) and 50 μm (A-D, right microphotograph).

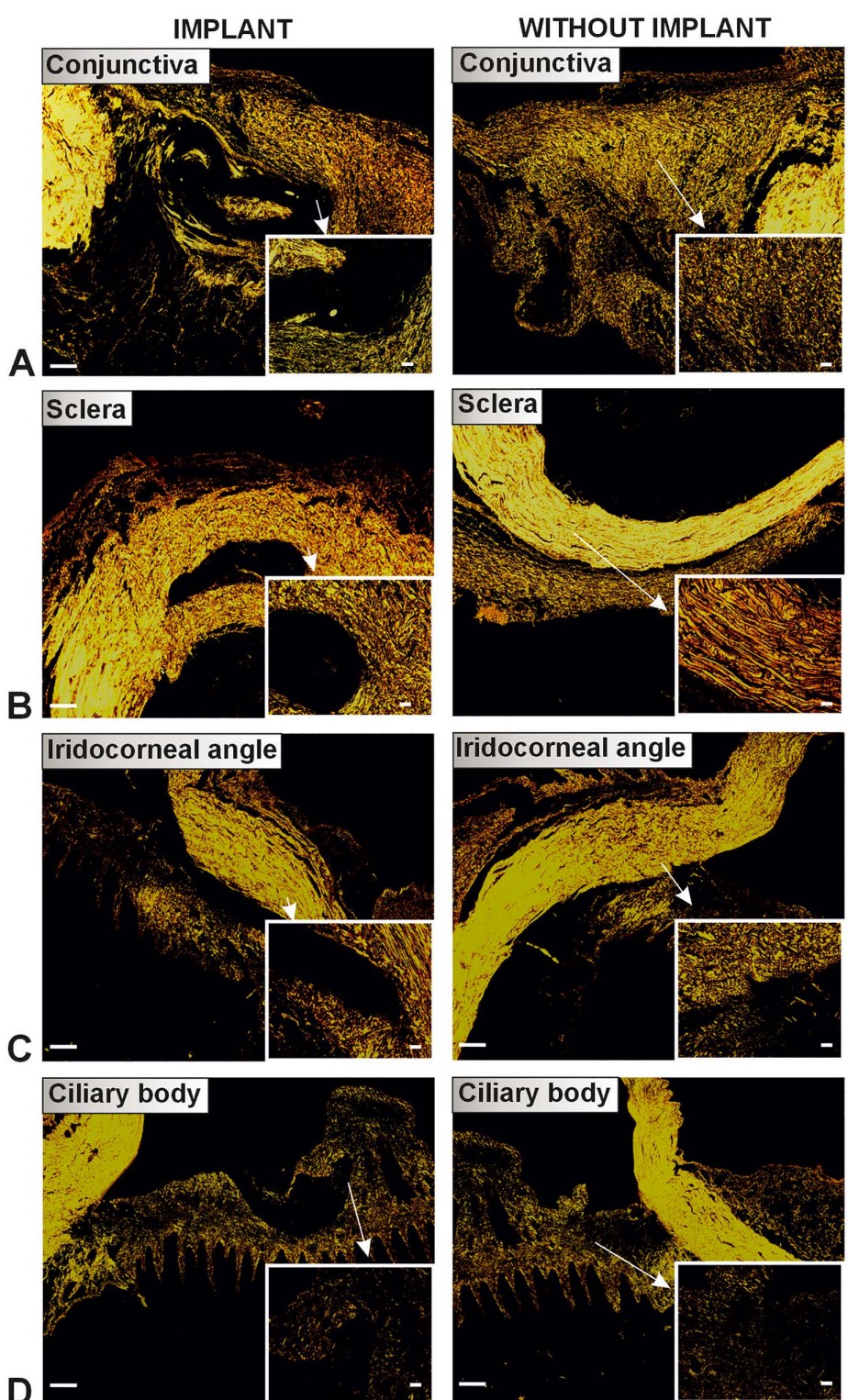

**Fig 10. Comparison of collagen distribution in the implant and without implant groups.** Scale bars 500 μm (A-D, left side) and 50 μm (A-D, right microphotograph).

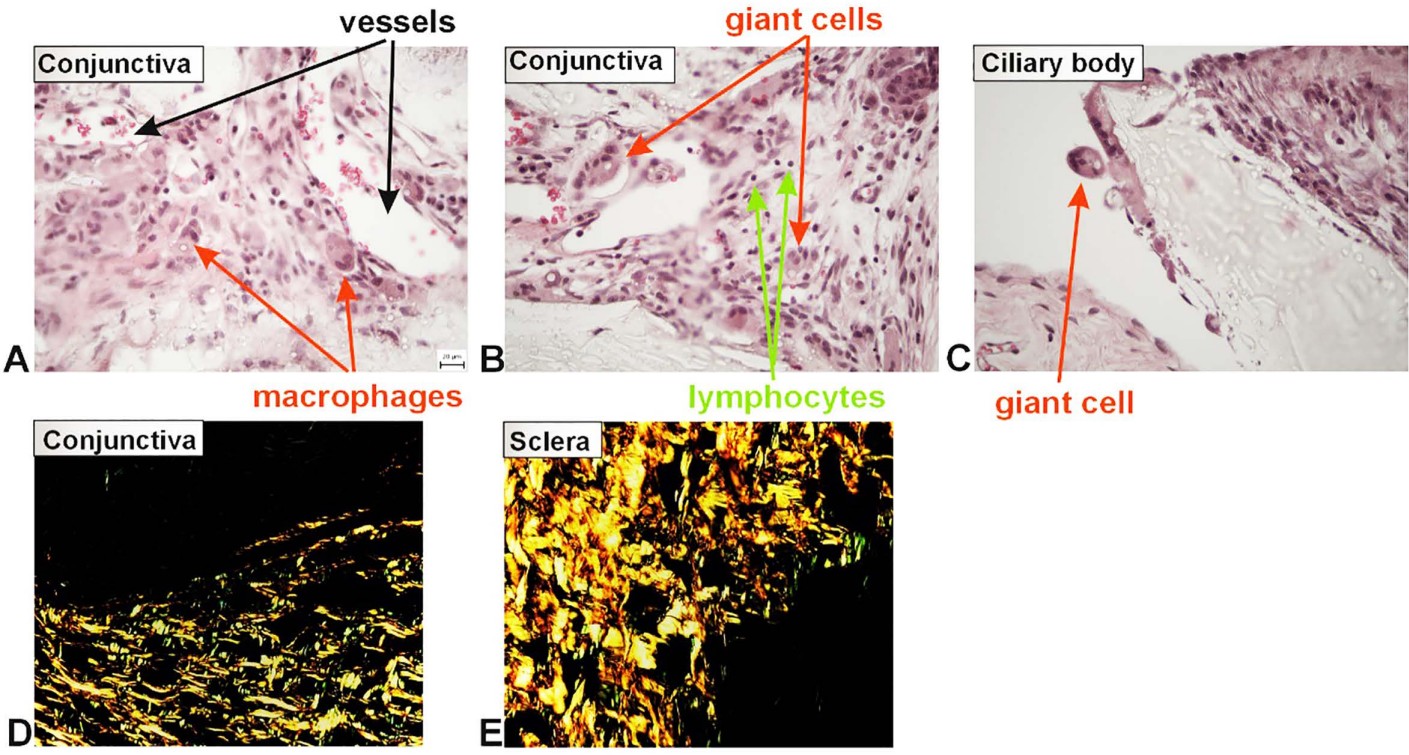

**Fig 11. Qualitative changes in the implant subgroup.**

A diverse array of biomaterials, including bioinert metals and synthetic polymers, has been utilized in the development of GDIs, with evaluations conducted in both preclinical models and clinical trials [23–25]. However, variability in post-operative outcomes across these material types underscores the ongoing need for alternative, material-focused strategies aimed at mitigating biomaterial-associated fibrosis.

Fibroblast responses from scleral, conjunctival, and Tenon's capsule tissues to fibrotic stimuli have been extensively characterized [26–28]. In particular, Tenon's fibroblasts have been shown to exhibit well-defined vascular endothe-lial growth factor (VEGF) production and proliferative responses *in vitro* when exposed to fibrotic signals and various small-molecule drugs [29]. Moreover, subsets of conjunctival and scleral fibroblasts possess immunomodulatory proper-ties and play active roles in tissue remodeling and collagen synthesis. Notably, scleral fibroblasts display a pronounced pro-fibrotic response upon exposure to activating factors such as transforming growth factor beta (TGF-β) [30].

Josyula et al. proposed that a nanofibrous surface would better replicate the structural and mechanical characteristics of healthy extracellular matrix and would, therefore, be less likely to elicit a foreign body response from fibroblasts com-pared to smooth surface GDIs [29]. In both *in vitro* and *in vivo* (rabbit eyes) studies, the nanofiber architecture was found to suppress fibroblast activation and promote cellular integration. Furthermore, GDIs with a nanofiber surface resulted in reduced subconjunctival fibrosis in rabbit eyes [31].

The results from our study confirm this statement, given the distribution of the fraction of total collagen and type I colla-gen in the implant subgroup.

Sacchi et al. considered two opposite models of fibrosis: skin and lung fibrosis [32]. Skin fibrosis typically follows a self-limiting course triggered by inflammatory signals. In contrast, pulmonary fibrosis represents a relentless and progressive form of scar-ring, sustained mainly by the combined effects of cellular senescence and apoptosis. The skin and ocular surface share a close

biological relationship, reflected by their overlapping immunobiological features, steroidogenic activity, and responses to allergic challenges. From an embryological perspective, the conjunctiva can be considered a specialized modification of skin. Fibrosis of the conjunctiva is primarily mediated by mast cells, growth factors such as TGFβ1, and M2-like macrophages, elements that also play central roles in skin fibrosis. Mast cells contribute through the release of cytokines that attract neutrophils and monocytes, chymase and tryptase that enhance fibroblast proliferation and their transformation into myofibroblasts, and histamine that stimulates collagen I and III synthesis as well as growth factor production [32]. Our findings support the observations made by Sacchi et al. regarding the biological parallels between the skin and the conjunctiva. A chronic inflammatory response was evident around the implant, characterized by the presence of newly formed collagen fibers, predominantly type III.

Limitations that apply to this study include a relatively short follow-up period and a relatively small sample size.

## Conclusion

For easier implantation, minor technical adjustments like implant narrowing and scleral fixation of the GDI were done in *in vitro* experiments. Our *in vivo* study showed that implantation of our unique nanofiber glaucoma device was safe (within the time limits of the follow-up) and technically very feasible. No serious perioperative or postoperative complications were observed; however, there was one scleral extrusion of the device, which we attribute to insufficient conjunctival fixation. A statistically significant IOP reduction was achieved in the study group. Further studies into the long-term effectiveness of the implant, including extended follow-up and evaluation in combination with other surgical approaches, e.g., cataract surgery paired with nanofiber device implantation, would be beneficial.

## Author contributions

**Conceptualization:** Adela Klezlova, Petr Bulir, Andrea Sidova, Martina Grajciarova, Lenka Vankova, Zbynek Tonar, Pavel Studeny.

**Data curation:** Adela Klezlova, Magdalena Netukova, Jana Vranova, Katarina Urbaniova, Martina Grajciarova.

**Formal analysis:** Adela Klezlova, Andrea Sidova, Jana Vranova.

**Investigation:** Petr Bulir, Martina Grajciarova, Lenka Vankova.

**Methodology:** Adela Klezlova, Pavel Studeny.

**Project administration:** Zbynek Tonar, Pavel Studeny.

**Supervision:** Alexandr Stepanov, Zbynek Tonar.

**Writing – original draft:** Adela Klezlova, Andrea Sidova, Martina Grajciarova, Lenka Vankova.

**Writing – review & editing:** Alexandr Stepanov, Zbynek Tonar, Pavel Studeny.

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
