## [Decision Letter · Decision Letter 0]

13 Aug 2025

Dear Dr. Stepanov,

Thank you for submitting your manuscript to PLOS ONE. After careful consideration, we feel that it has merit but does not fully meet PLOS ONE’s publication criteria as it currently stands. Therefore, we invite you to submit a revised version of the manuscript that addresses the points raised during the review process.

We look forward to receiving your revised manuscript.

Kind regards,

Yalong Dang

Academic Editor

PLOS ONE

Journal Requirements:

2. To comply with PLOS One submissions requirements, in your Methods section, please provide additional information regarding the experiments involving animals and ensure you have included details on (1) methods of sacrifice, (2)  efforts to alleviate suffering.

4. Please include your tables as part of your main manuscript and remove the individual files. Please note that supplementary tables (should remain/ be uploaded) as separate "supporting information" files.

6. Thank you for stating the following financial disclosure:

“The study received support from the Charles University Cooperatio Program, research area MED/DIAG, the grant SVV – 2025 No 260 773, and from the Ministry of Education, Youth and Sports under the project FIND No. CZ.02.1.01/0.0/0.0/16_019/0000787, UNCE/MED006 Center of Excellence (Charles University), and AZV, grant number: NU 23-08-00586.”

7. Thank you for stating the following in the Acknowledgments Section of your manuscript:

“The study received support from the Charles University Cooperatio Program, research area MED/DIAG, the grant SVV – 2025 No 260 773, and from the Ministry of Education, Youth and Sports under the project FIND No. CZ.02.1.01/0.0/0.0/16_019/0000787, UNCE/MED006 Center of Excellence (Charles University), and AZV, grant number: NU 23-08-00586.”

“The study received support from the Charles University Cooperatio Program, research area MED/DIAG, the grant SVV – 2025 No 260 773, and from the Ministry of Education, Youth and Sports under the project FIND No. CZ.02.1.01/0.0/0.0/16_019/0000787, UNCE/MED006 Center of Excellence (Charles University), and AZV, grant number: NU 23-08-00586.”

9. We notice that your supplementary figures are uploaded with the file type 'Other'. Please amend the file type to 'Supporting Information'. Please ensure that each Supporting Information file has a legend listed in the manuscript after the references list.

Reviewers' comments:

Reviewer's Responses to Questions

**Comments to the Author**

1. Is the manuscript technically sound, and do the data support the conclusions?

Reviewer #1: Partly

Reviewer #2: Partly

Reviewer #3: Yes

2. Has the statistical analysis been performed appropriately and rigorously?

Reviewer #1: Yes

Reviewer #2: Yes

Reviewer #3: Yes

3. Have the authors made all data underlying the findings in their manuscript fully available?

Reviewer #1: Yes

Reviewer #2: Yes

Reviewer #3: Yes

4. Is the manuscript presented in an intelligible fashion and written in standard English?

Reviewer #1: Yes

Reviewer #2: Yes

Reviewer #3: No

Reviewer #1: The study is interesting and case controlled, however some changes are necessary before it can be considered for publication.

Intro: how did you get to synthetic polymer polyvinylidenefluoride (PVDF)? there are previous studies/experiences on this materials? did you mention hernia meshes, but the aim in glaucoma surgery compared to hernia repair is different, as in eye surgery we don't want to promote any healing process, compared to hernia repair.

the device is not properly described and reader cannot really figure out what the study is really talking about. The device design (and the work behind the design) needs to be properly explained.

Is the device done for trabecular meshwork outflow or for subconj filtration? It is not clear.

This point is important: "The minimal cellular attachment and proliferation suggest that the material effectively inhibits fibroblast colonization, a critical requirement for minimizing fibrotic encapsulation in glaucoma drainage systems", how did you explain this property?

1 months is a short time period for the histological analysis, as tissue changes require weeks and months. I would mention it among the study limitation. Consider to mention this work for a deep analysis of the bleb fibrosis and consider to comment your results compared to the review results (Fighting Bleb Fibrosis After Glaucoma Surgery: Updated Focus on Key Players and Novel Targets for Therapy. Sacchi M et all.)

Device design is not clear, I would work on improving the device design.

Minor comments

abstract, methods: I would spend few words explaining what new nanofibers GDI are

line 59 typos error: to groups

line 69: "did not mention", maybe I would change in "did not record"

line 175 typos error: reductionby

intro: "The drawback, however, is that their use may lead to undesirable side effects,

such as cardiac or respiratory complications." cardiac or respiratory complications are just for BB; however they are not the most common drawback of topical medication (they are, from my point of view, glaucoma-related OSD and low adherence).

line 181: "Laser therapy can partially reduce the IOP, but without long-term effect" The LiGHT study has shown that the effect of SLT can indeed last for a long time.

later: "However, it may be an alternative for patients who are irresponsible or cannot use medication (5)." SLT it is actually considered as first step procedure, as for EGS guidelines.

In conclusion,

i) I feel we cannot draw any conclusion about the efficacy of the device, or the safety profile, as this is a in vitro and in vivo study with small sample and short follow-up. I think it is need to be addressed. I do think the value of this work are: i) to shine a light on a new material, potential suitable for eye surgery, ii) to prove the biocompatibility of this material iii) to propose a new design for glaucoma drainage.

ii) intro should be sharper, and really and briefly introduce to the work. Consider to rephrase and maybe shorten it

iii) check and update the reference, when necessary

Reviewer #2: General Assessment

The manuscript investigates the biological and clinical performance of new nanofiber GDI. The topic is relevant to the field of biomaterials and tissue engineering, and the experimental design appears to follow established protocols. However, the manuscript contains several critical shortcomings in methodology, data presentation, and interpretation. These issues must be addressed to ensure scientific rigor, reproducibility, and clarity.

Major Comments

1. Missing Figure Captions and Legends

The manuscript lacks figure captions and detailed legends. This omission significantly impairs the reader’s ability to interpret visual data. Each figure should be accompanied by a comprehensive caption describing the experimental conditions, observed results, and any relevant statistical analysis. Figures should also be clearly referenced and discussed within the main text.

2. Incomplete Methodological Details

Several key aspects of the methodology are insufficiently described:

- The technique used for cell seeding onto the scaffolds is not specified.

- The rationale for selecting a seeding density of 1×10⁴ cells per well is not discussed.

- Only two time points (day 1 and day 8) are used; intermediate time points could provide more insight into cell-scaffold interactions.

- The staining method for cell nuclei is not mentioned.

- The MATLAB-based image analysis lacks detail regarding the algorithm, thresholding, and validation.

3. Absence of Biological Control Groups

No control groups are included to validate the biological performance of the scaffolds. Wells without scaffolds or with known biocompatible materials should be used as controls to ensure meaningful comparisons.

4. Lack of Proper Surgical Controls

The study design does not include a group in which the surgical procedure is performed without GDI implantation. This limits the ability to isolate biological responses caused solely by the scaffold. A sham-operated control group undergoing the same surgical steps without material placement is essential for distinguishing scaffold-specific effects from surgical trauma.

5. Inappropriate Histological Comparison

Histological comparisons between control and experimental eyes are questionable, as the control eyes did not undergo any surgical intervention. Without matching surgical conditions, histological differences may reflect procedural effects rather than material performance. This undermines the validity of tissue-level conclusions.

6. Absence of Technical Illustration of the GDI Scaffold

The manuscript does not include a figure or schematic detailing the technical specifications of the GDI scaffold. A visual representation showing geometry, porosity, dimensions, and material composition would enhance clarity and help readers understand the scaffold’s physical characteristics.

7. Referencing and Reproducibility Issues

Additionally, reagent sources are listed without catalog numbers, which limits reproducibility.

8. Language and Technical Accuracy

The manuscript contains grammatical errors and typographical issues (e.g., “cell qu antification”) that detract from its professionalism. A thorough language revision is recommended, ideally with assistance from a native English speaker or professional editor.

9. Limited Interpretation in the Conclusion Section

The conclusion focuses only on clinical findings and does not incorporate cellular or histological results. A more integrated discussion linking clinical outcomes with cellular responses would strengthen the manuscript and provide a more comprehensive understanding of the scaffold’s biological performance.

Minor Comments

- Define all abbreviations upon first use (e.g., DMEM, FBS, MTT).

- Clarify the spectrophotometric analysis parameters (e.g., wavelength, calibration).

- Consider adding a brief discussion of study limitations and future research directions.

Reviewer #3: In this study the authors report the effectiveness, postoperative complications, and histopathological findings of a new nanofiber glaucoma drainage implant (GDI) and observed promising results. This is a well-conducted study but a poorly written manuscript which is too lengthy and not easy to follow. Some sentences and explanations are repeated several times in the text and probably in the tables (I don’t see tables and figure legends in the uploaded file). I suggest rewriting the manuscript and avoid unnecessary or duplicated parts.

Below please find some comments to improve the manuscript.

Line 314:” without implant group was created from the intact parts of the same eyes”.

Please explain why fellow eyes were not used as control for histological study.

Line 349: “right eyes (eyes which underwent the both surgical procedures, i.e. with and without GDI) and left eyes (eyes without any surgery). This is confusing. As mentioned under methods, in 14 rabbits one eye received GDI and the other served as control with no surgery.

Line 355: “difference between preoperative IOP and IOP one day, one, two, three weeks, and one month after surgery.” This sentence has been repeated several times. The whole manuscript is too lengthy with too many details.

Line 361: “the implant and without implant groups”. Use a single name for the study groups such as the study and control groups throughout the manuscript.

Line 371: “The fabricated nanofibrous layer exhibited an area weight of approximately 20 g/m², corresponding to a thickness of around 100 µm.” while in line 275 it is read “After the several surgical tries of three different thickness of the implant (50, 100, 150 µm) we decided for the one with 150 µm”.

Lines 428-429: provide the exact p values.

Line 433: under results only provide the observed results without any interpretations (it tells us or it means…)which should be mentioned and discussed under discussion.

Discussion is too lengthy with irrelevant mention of MIGS or Cypass withdrawal starting from line 469.

I don’t see tables and figure legends in the uploaded file.

The manuscript needs professional English editing.

**Do you want your identity to be public for this peer review?** For information about this choice, including consent withdrawal, please see our Privacy Policy

Reviewer #1: No

Reviewer #2: **Yes: ** Tulay Simsek

Reviewer #3: No

---

## [Author Response · Author response to Decision Letter 1]

26 Sep 2025

Reviewers' comments:

Reviewer #1: The study is interesting and case controlled, however some changes are necessary before it can be considered for publication.

Intro: how did you get to synthetic polymer polyvinylidenefluoride (PVDF)? there are previous studies/experiences on this materials?

Thank you very much for your question. We have considerable experience with the synthetic polymer PVDF in the fabrication of nanofiber materials for various applications, including biomedical use. Its specific application in the treatment of glaucoma has been previously described in the publication “A PVDF electrospun antifibrotic composite for use as a glaucoma drainage implant,” which has been presented at several conferences. In this context, the material was characterized with respect to its in vitro, ex vivo, and in vivo behavior, which involved extensive laboratory studies focused on flow characteristics.

did you mention hernia meshes, but the aim in glaucoma surgery compared to hernia repair is different, as in eye surgery we don't want to promote any healing process, compared to hernia repair.

Thank you for this important remark. We agree that the clinical objectives of glaucoma surgery and hernia repair differ substantially. In hernia repair, the primary purpose of the mesh is to provide mechanical reinforcement and simultaneously support the healing process. In contrast, in glaucoma surgery, excessive healing and scarring are undesirable since these processes compromise aqueous humor outflow and long-term surgical success. Our reference to hernia meshes was therefore not intended to suggest similarity in clinical goals, but rather to highlight the anti-adhesive properties of PVDF that have been recognized in this context. These properties are equally advantageous in glaucoma surgery, where minimizing fibrosis and adhesion formation is of key importance.

the device is not properly described and reader cannot really figure out what the study is really talking about. The device design (and the work behind the design) needs to be properly explained.

Is the device done for trabecular meshwork outflow or for subconj filtration? It is not clear.

We appreciate the reviewer’s comment and agree that a more detailed description of the device is essential for clarity. In the revised manuscript, we have expanded the section describing the design and functionality of the implant, including the rationale behind its development.

This point is important: "The minimal cellular attachment and proliferation suggest that the material effectively inhibits fibroblast colonization, a critical requirement for minimizing fibrotic encapsulation in glaucoma drainage systems", how did you explain this property?

Thank you for highlighting this important aspect. These properties have indeed been corroborated through multiple in vitro investigations. PVDF's hydrophobic nature fundamentally impairs cell adhesion and proliferation because mammalian cells typically require a moderately hydrophilic surface that supports adsorption of adhesion-mediating proteins such as fibronectin. These conditions are not favored by highly hydrophobic polymers (DOI: 10.1002/jbm.b.30638 ). Moreover, unmodified PVDF nanofibers exhibit significant hydrophobicity (with water contact angles around 124°), which correlates with reduced cell-surface interactions (https://doi.org/10.3390/polym17030330). These findings are consistent with our own observations, i.e., the hydrophobic character of the PVDF material likely deters fibroblast colonization, thereby helping to minimize fibrotic encapsulation. In the revised manuscript, we have now emphasized this mechanistic explanation, along with relevant citations to strengthen the argument.

1 months is a short time period for the histological analysis, as tissue changes require weeks and months. I would mention it among the study limitation. Consider to mention this work for a deep analysis of the bleb fibrosis and consider to comment your results compared to the review results (Fighting Bleb Fibrosis After Glaucoma Surgery: Updated Focus on Key Players and Novel Targets for Therapy. Sacchi M et all.)

Sacchi et al. considered two opposite models of fibrosis: skin and lung fibrosis. Skin fibrosis typically follows a self-limiting course triggered by inflammatory signals. In contrast, pulmonary fibrosis represents a relentless and progressive form of scarring, sustained mainly by the combined effects of cellular senescence and apoptosis. The skin and ocular surface share a close biological relationship, reflected by these overlapping immunohistological features, steroidogenic activity, and responses to allergic challenges. From an embryological perspective, the conjunctiva can be considered a specialized modification of the skin. Fibrosis of the conjunctiva is primarily mediated by mast cells, growth factors such as TGF β1, and M2-like macrophages; elements that also play central roles in skin fibrosis. Mast cells contribute through the release of cytokines that attract neutrophils and monocytes, chymase and tryptase that enhance fibroblast proliferation and their transformation into myofibroblasts, and histamine that stimulates collagen I and III synthesis as well as growth factor production. In our study, we confirmed the ideas of Sacchi et al. regarding the biological relationship between the skin and the conjunctiva. A chronic inflammatory reaction with newly formed collagen fibers, especially type III, was present around the implant.

Device design is not clear, I would work on improving the device design.

We added two figures, showing insertion of the GDI in vitro via a scleral tunnel. Hopefully, it better describes the device and its appearance during implantation.

Fig. 2 A: In vitro GDI inserted via a scleral tunnel; B: GDI bent and covered by the conjunctiva

Minor comments

abstract, methods: I would spend few words explaining what new nanofibers GDI are

Added to the revised text.

line 59 typos error: to groups

Revised

line 69: "did not mention", maybe I would change in "did not record"

Revised

line 175 typos error: reduction by

Revised

intro: "The drawback, however, is that their use may lead to undesirable side effects, such as cardiac or respiratory complications." cardiac or respiratory complications are just for BB; however they are not the most common drawback of topical medication (they are, from my point of view, glaucoma-related OSD and low adherence).

Revised

line 181: "Laser therapy can partially reduce the IOP, but without long-term effect" The LiGHT study has shown that the effect of SLT can indeed last for a long time. later: "However, it may be an alternative for patients who are irresponsible or cannot use medication (5)." SLT it is actually considered as first step procedure, as for EGS guidelines.

Revised

In conclusion,

i) I feel we cannot draw any conclusion about the efficacy of the device, or the safety profile, as this is a in vitro and in vivo study with small sample and short follow-up. I think it is need to be addressed. I do think the value of this work are: i) to shine a light on a new material, potential suitable for eye surgery, ii) to prove the biocompatibility of this material iii) to propose a new design for glaucoma drainage.

ii) ii) intro should be sharper, and really and briefly introduce to the work. Consider to rephrase and maybe shorten it

iii) check and update the reference, when necessary

Revised

Reviewer #2: General Assessment

The manuscript investigates the biological and clinical performance of new nanofiber GDI. The topic is relevant to the field of biomaterials and tissue engineering, and the experimental design appears to follow established protocols. However, the manuscript contains several critical shortcomings in methodology, data presentation, and interpretation. These issues must be addressed to ensure scientific rigor, reproducibility, and clarity.

Major Comments

1. Missing Figure Captions and Legends

The manuscript lacks figure captions and detailed legends. This omission significantly impairs the reader’s ability to interpret visual data. Each figure should be accompanied by a comprehensive caption describing the experimental conditions, observed results, and any relevant statistical analysis. Figures should also be clearly referenced and discussed within the main text.

Figure captions and detailed legends were added.

2. Incomplete Methodological Details

Several key aspects of the methodology are insufficiently described:

- The technique used for cell seeding onto the scaffolds is not specified.

- The rationale for selecting a seeding density of 1×10⁴ cells per well is not discussed.

- Only two time points (day 1 and day 8) are used; intermediate time points could provide more insight into cell-scaffold interactions.

- The staining method for cell nuclei is not mentioned.

- The MATLAB-based image analysis lacks detail regarding the algorithm, thresholding, and validation.

Thank you for your comments. Here are the answers to your questions.

• The detailed procedure for cell seeding onto the scaffolds is provided in the In Vitro Characterization section and includes all the necessary information to reproduce the experiment.

• Seeding the scaffolds with cells at a density of 1×10⁴ is consistent with our standard laboratory protocols, which are documented in our previous publications. At this specific density, the cells typically exhibit optimal attachment and proliferation while preventing premature confluence, which is why it is commonly used.

• We agree that additional time points, for example, after 2 days, could be included; however, considering the intended application, namely GDI, the most relevant information is provided by in vitro testing at 1 and 8 days of culture. Based on our experience, intermediate data does not offer any additional significant insights.

• The method for cell staining and cell counting in MATLAB is described in Horakova’s publication, as noted in the last sentence of the paragraph; however, it has now been included in the text for clarity.

3. Absence of Biological Control Groups

No control groups are included to validate the biological performance of the scaffolds. Wells without scaffolds or with known biocompatible materials should be used as controls to ensure meaningful comparisons.

4. Lack of Proper Surgical Controls

The study design does not include a group in which the surgical procedure is performed without GDI implantation. This limits the ability to isolate biological responses caused solely by the scaffold. A sham-operated control group undergoing the same surgical steps without material placement is essential for distinguishing scaffold-specific effects from surgical trauma.

5. Inappropriate Histological Comparison

Histological comparisons between control and experimental eyes are questionable, as the control eyes did not undergo any surgical intervention. Without matching surgical conditions, histological differences may reflect procedural effects rather than material performance. This undermines the validity of tissue-level conclusions.

Thank you for your comments. We decided to compare the study group with GDI and the control group without any surgical procedure, as ethical concerns prohibited us from performing surgery on both eyes of one animal to avoid the risk of complete blindness. However, we believe the comparison is still valid.

6. Absence of Technical Illustration of the GDI Scaffold

The manuscript does not include a figure or schematic detailing the technical specifications of the GDI scaffold. A visual representation showing geometry, porosity, dimensions, and material composition would enhance clarity and help readers understand the scaffold’s physical characteristics.

The porosity of the GDI was determined in accordance with ASTM F316 using a capillary flow porometer (Porometer 3G, Quantachrome, USA). The measured porosity was 85.62 ± 2.85%.

For better understanding of the appearance of the GDI and its function we added two more pictures, where is the GDI shown during its in vitro implantation (Fig. 2 A and B). We believe that twoo other pictures and their descriptions (see please above) could help the readers to understand the GDI and the surgical procedure.

7. Referencing and Reproducibility Issues

Additionally, reagent sources are listed without catalog numbers, which limits reproducibility.

8. Language and Technical Accuracy

The manuscript contains grammatical errors and typographical issues (e.g., “cell qu antification”) that detract from its professionalism. A thorough language revision is recommended, ideally with assistance from a native English speaker or professional editor.

9. Limited Interpretation in the Conclusion Section

The conclusion focuses only on clinical findings and does not incorporate cellular or histological results. A more integrated discussion linking clinical outcomes with cellular responses would strengthen the manuscript and provide a more comprehensive understanding of the scaffold’s biological performance.

Minor Comments

- Define all abbreviations upon first use (e.g., DMEM, FBS, MTT).

- Clarify the spectrophotometric analysis parameters (e.g., wavelength, calibration).

- Consider adding a brief discussion of study limitations and future research directions.

Reviewer #3:

In this study the authors report the effectiveness, postoperative complications, and histopathological findings of a new nanofiber glaucoma drainage implant (GDI) and observed promising results. This is a well-conducted study but a poorly written manuscript which is too lengthy and not easy to follow. Some sentences and explanations are repeated several times in the text and probably in the tables (I don’t see tables and figure legends in the uploaded file). I suggest rewriting the manuscript and avoid unnecessary or duplicated parts.

Below please find some comments to improve the manuscript.

Line 314:” without implant group was created from the intact parts of the same eyes”. Please explain why fellow eyes were not used as control for histological study.

Thank you for your comment. To minimize variability between individual eyes, we compared regions of the same eye with and without the implant, thereby obtaining a direct internal control within the same biological environment.

Line 349: “right eyes (eyes which underwent the both surgical procedures, i.e. with and without GDI) and left eyes (eyes without any surgery). This is confusing. As mentioned under methods, in 14 rabbits one eye received GDI and the other served as control with no surgery.

Revised

Line 355: “difference between preoperative IOP and IOP one day, one, two, three weeks, and one month after surgery.” This sentence has been repeated several times. The whole manuscript is too lengthy with too many details.

Revised

Line 361: “the implant and without implant groups”. Use a single name for the study groups such as the study and control groups throughout the manuscript.

Revised

Line 371: “The fabricated nanofibrous layer exhibited an area weight of approximately 20 g/m², corresponding to a thickness of around 100 µm.” while in line 275 it is read “After the several surgical tries of three different thickness of the implant (50, 100, 150 µm) we decided for the one with 150 µm”.

Revised in whole paper

Lines 428-429: provide the exact p values.

Revised

Line 433: under results only provide the observed results without any interpretations (it tells us or it means…) which should be mentioned and discussed under discussion. Discussion is too lengthy with irrelevant mention of MIGS or Cypass withdrawal starting from line 469.

The Discussion was changed accordingly.

I don’t see tables and figure legends in the uploaded file. The manuscript needs professional English editing.

A native speaker has revised the whole manuscript.

6. PLOS authors have the option to publish the peer review history of their article (what does t

---

## [Decision Letter · Decision Letter 1]

16 Oct 2025

New nanofiber glaucoma drainage implant: effectiveness, safety, first in vivo results and optimization of surgical technique

PONE-D-25-39325R1

Dear Dr. Stepanov,

We’re pleased to inform you that your manuscript has been judged scientifically suitable for publication and will be formally accepted for publication once it meets all outstanding technical requirements.

Kind regards,

Yalong Dang

Academic Editor

PLOS ONE

Additional Editor Comments (optional):

Reviewers' comments:

Reviewer's Responses to Questions

**Comments to the Author**

Reviewer #2: All comments have been addressed

Reviewer #3: All comments have been addressed

2. Is the manuscript technically sound, and do the data support the conclusions?

Reviewer #2: Yes

Reviewer #3: Yes

3. Has the statistical analysis been performed appropriately and rigorously?

Reviewer #2: Yes

Reviewer #3: Yes

4. Have the authors made all data underlying the findings in their manuscript fully available?

Reviewer #2: Yes

Reviewer #3: Yes

5. Is the manuscript presented in an intelligible fashion and written in standard English?

Reviewer #2: Yes

Reviewer #3: Yes

Reviewer #2: Dear Authors, The revisions have been appropriately and satisfactorily completed. I commend your efforts and wish you continued success in your scholarly work.

Reviewer #3: (No Response)

**Do you want your identity to be public for this peer review?** For information about this choice, including consent withdrawal, please see our Privacy Policy

Reviewer #2: **Yes: ** Tulay Simsek Tülay ŞİMŞEK, MD Professor in Ophthalmology

Reviewer #3: No

---

## [Editor Report · Acceptance letter]

PONE-D-25-39325R1

PLOS ONE

Dear Dr. Stepanov,

I'm pleased to inform you that your manuscript has been deemed suitable for publication in PLOS ONE. Congratulations! Your manuscript is now being handed over to our production team.

Kind regards,

on behalf of

Dr Yalong Dang

Academic Editor

PLOS ONE